

# Preoperative prediction of p16 expression in cervical cancer by amide proton transfer imaging combined with diffusion kurtosis imaging: a retrospective study

XingChen Wu[1], Chen Xu[2], Xiaoyan Zhang[3], Qianqian Qu[3], Zhe Wang[3] and Kai Deng[3]

[1] School of Medical Imaging, Shandong Second Medical University, Weifang, Shandong Province, China
[2] Clinical Medical College, Jining Medical University, Jining, Shandong Province, China
[3] Department of Radiology, The First Affiliated Hospital of Shandong First Medical University & Shandong Provincial Qianfoshan Hospital, Jinan, Shandong Province, China

Corresponding author
Kai Deng, 289954749@qq.com

## ABSTRACT

**Background:** Immunohistochemical p16 expression is important for the assessment of risk factors and prognosis of cervical cancer. This study evaluated the correlation between preoperative amide proton transfer imaging (APT), diffusion kurtosis imaging (DKI), and p16 expression in cervical cancer.

**Methods:** Fifty-five records of patients with cervical cancer with preoperative magnetic resonance imaging (MRI) and complete postoperative pathology were retrospectively analyzed. Uterine MRI scans were examined, and the corresponding APT values, mean kurtosis (MK), and mean diffusivity (MD) were obtained. Pathology was used to determine p16 positivity. The predictive effects of the APT- and DKI-derived parameters applied alone or in combination with p16 expression were compared.

**Results:** Among the 55 cases of cervical cancer, 35 were p16 positive and 20 were p16 negative. The receiver operating characteristic curve area (AUC) of p16 was 0.809, 0.801, 0.790, 0.845, 0.866, and 0.871 when APT, MK, and MD were used individually and then jointly. The effect of APT combined with the DKI derivative parameters was more pronounced than when they were used separately.

**Conclusion:** Preoperative APT and DKI imaging are valuable in predicting the expression of p16 in cervical cancer, and the combination of the two can improve the prediction efficacy. These findings could help to guide treatment decisions and determine prognosis in cervical cancer.

# INTRODUCTION

Cervical cancer is the fourth most common cancer among women worldwide, which mainly affects low- and middle-income countries. The main preventable risk factors for cervical cancer include human papillomavirus (HPV) infection, multiple sex partners, smoking, immunocompromised status (*e.g.*, AIDS), chlamydia infection, long-term use of

oral contraceptives, having multiple full-term pregnancies, young age at first full-term pregnancy, low economic status, and low consumption offruits and vegetables (*Brisson et al., 2020*). Despite preventive measures, such as cervical cancer screening, the mortality rate from cervical cancer remains high.

The tumor suppressor p16 is overexpressed in cervical intraepithelial neoplasia and many invasive cervical cancers (*McLaughlin-Drubin, Park & Munger, 2013*; *Liao et al., 2014*). It is a G1/S phase cell cycle inhibitor that prevents cell replication by binding to the cell cycle protein cyclen D1-cyclen-dependent kinase 4/6 complex and inhibits retinoblastoma (Rb) protein phosphorylation. Therefore, p16 promotes the formation of the Rb-E2F inhibitory complex, which in turn blocks the G1/S progression of the cell cycle (*Romagosa et al., 2011*; *Calil et al., 2014*).

Presently, p16 is commercially approved as an immunomarker for surgical pathology material such as diagnostic and excisional biopsies, p16 staining is highly specific for squamous abnormalities and shows a high degree of concordance among pathologists. Many studies exist on the role of p16 in cervical cancer, finding that p16 expression predicted a favorable prognosis in specific subgroups of patients. Although a number of these studies suggested a close correlation between p16 expression and prognosis, some of the results were not significant. However, p16 expression is important for the assessment of risk factors and prognosis of cervical cancer (*da Mata et al., 2021*).

The latest International Federation of Gynecology and Obstetrics and European Society of Urogenital Radiology systems have focused on magnetic resonance imaging (MRI) to accurately measure tumor size and characterize deep mesenchymal infiltration and parametric involvement. Multimodal MRI combines multiple imaging techniques providing more comprehensive and accurate diagnostic information. Although conventional MRI can provide a morphological assessment of cervical cancer, visualizing the microstructure of a tumor without specific indicators is difficult. Different advanced MRI modalities, including molecular imaging, can provide additional complementary information for clinical diagnosis and prognostic assessment, optimizing therapeutic outcomes (*Manganaro et al., 2021*). Diffusion kurtosis imaging (DKI) can reflect the diffusion status of water molecules in the tissue (*Iima & Le Bihan, 2016*); and amide proton transfer (APT) imaging can detect free proteins and peptide molecules located in the cell matrix, thus reflecting the metabolic and pathophysiological information inside the cells (*Sheng et al., 2024*).

In 2018, *Takayama et al. (2018)* reported the first use of 2D APT imaging on the uterus. The APT signal intensity was positively correlated with the histologic grades of endometrioid endometrial adenocarcinoma (*Nishie et al., 2018*), later, a published APT study on 31 cervical squamous cell carcinoma patients showed that APT signal intensity was positively correlated with histological grades (*Takayama et al., 2018*). The application of DKI in cervical cancer is still in the research stage, but it shows great potential in evaluating tumor microenvironment, differential diagnosis and curative effect detection (*Wang et al., 2020*). Studies that combine these two to predict p16 expression in cervical cancer are rare.

This study investigated the correlation of APT/DKI parameter characteristics of preoperative MR functional sequences and the histochemical expression of p16 in cervical cancer to better inform treatment selection and to help determine prognosis.

## MATERIALS AND METHODS

### Study subjects

Fifty-five patients with cervical cancer treated at our hospital between May 2022 and October 2024 were retrospectively included. The inclusion criteria were as follows: (1) postoperative pathologically confirmed cervical cancer, (2) history of pelvic MRI 1–2 weeks before treatment, and (3) puncture biopsy or surgical treatment with routine pathological examination and immunohistochemical testing within 2 weeks after MRI. (4) Surgical treatment as the original treatment, and (5) no radiotherapy before surgery. The exclusion criteria were as follows: (1) preoperative treatment for cervical lesions; (2) incomplete information, such as missing pathology immunohistochemistry testing; (3) unclear imaging findings (breathing, motion, or metal artifacts) that interfered with the observation and measurement of lesions; and (4) other malignancies. The study protocol was approved by the Medical Ethics Committee of our hospital (Medical Ethics Committee of the First Affiliated Hospital of Shandong First Medical University & Shandong Provincial Qianfoshan Hospital, 2021S849). Informed consents were confirmed verbally.

### Methods of study

#### Magnetic resonance examination

The patients were instructed by their physicians to ingest an appropriate amount of water to moderately fill their bladders before the examination. A Philips 3.0T MRI device (Ingenia CX; Philips Healthcare, Netherlands) with an 18-channel phased array coil was used. In the supine position, patients' feet were advanced and a scan was performed from the bilateral superior borders of the ilia to the femoral necks. Plain MRI was first performed, followed by APT and DKI sequence scanning. Detailed MRI parameters of APT and DKI were listed in Table 1.

#### Image analysis

Image analysis was performed independently by two radiologists with 5 and 7 years of diagnostic experience, who were blinded to the pathology results respectively, in pelvic MRI. APT images were reconstructed and transferred to a workstation (IntelliSpace Portal V9.0.4.31010) for subsequent analysis. APT-weighted images (APTw) were acquired using a pulsed CEST sequence with saturation frequency offsets centered at ±3.5 ppm. The APT signal was calculated using the magnetization transfer ratio asymmetry ($MTR_{asym}$) at 3.5 ppm:

$$MTR_{asym}(3.5 \text{ ppm}) = \frac{S(-3.5 \text{ ppm}) - S(3.5 \text{ ppm})}{S_0}$$

**Table 1 Detailed MRI parameters of APT and DKI.**

| Parameters | DKI | APT |
|---|---|---|
| Sequence/orientation | TSE/axial | TSE/axial |
| TR (ms) | 6,168 | 6,491 |
| TE (ms) | 58 | 8.5 |
| Field of view (mm$^2$) | 300 × 400 | 250 × 346 |
| Matrix | 120 × 158 | 140 × 192 |
| b-values (s/mm$^2$) | 0 50 100 800 1,500 2,000 | NA |
| Slice thickness (mm) | 5 | 5 |
| Fat suppression | SPAIR | SPAIR |
| Scanning time (min) | 7.4 | 6.5 |

**Note:**
  TSE, Turbo Spin Echo; TR, Repetition Time; TE, Echo Time.

where $S(-3.5$ ppm$)$ and $S(3.5$ ppm$)$ denote the signal intensities at corresponding offset frequencies, and $S_0$ represents the signal without saturation. The resulting APTw images reflect the endogenous mobile protein and peptide content.

The region of interest (ROI) of the APT image was first outlined on the T2W axial image by selecting the largest cross-section of the tumor, and then hand-drawn along the contour of the solid tumor to outline the region of interest (with the lesion edge preserved by 1–2 mm), and the ROIs were delineated according to the requirements of including the solid parts of the tumor and avoiding large blood vessels, calcification, cystic changes, and hemorrhage as much as possible. The APT image was then fused with the axial T2W image, the largest cross-section of the tumor was selected on the fused image, and the APT value was recorded.

The DKI images were imported into MR_Diffusion software in digital imaging and communications in medicine format and fitted to generate map plots of mean diffusivity (MD) and mean kurtosis (MK). DKI images were obtained using a multi-b-value diffusion-weighted sequence with at least three b-values (*e.g.*, 0, 800, 1,500 s/mm$^2$) in multiple diffusion directions. Kurtosis maps were derived by fitting the DKI model:

$$\ln\left(\frac{S(b)}{S_0}\right) = -bD + \frac{1}{6}b^2 D^2 K$$

where $D$ is the diffusion coefficient and $K$ is the kurtosis parameter. Fitting was conducted using. The ROIs of the DKIs were then sketched by the same two radiologists, and the related data of MD and MK were analyzed based on the ROIs. MRIcroGL software was used to save each sequence image as a Neuroimaging Informatics Technology Initiative file.

Three measurements were performed and averaged for each observer, and the final APT and DKI values were the average of the two observers.

### Immunohistochemical p16 test

Cervical tissue samples were obtained, fixed, and prepared into sections by a paraffin embedding technique. The proteins in the sections were exposed through steps for

degreasing and dissociation, p16INK4a antibody was added to the sections to bind to the p16 protein in the tissue samples. Appropriate labeling conjugates (*e.g.*, horseradish peroxidase, alkaline phosphatase) were used to form complexes with the antibody bound to the p16 protein, and the appropriate staining substrate was added to produce color in the enzyme reaction. The reaction time was adjusted according to the specific requirements of the laboratory. Microscopy was used to evaluate the expression of p16 protein based on the intensity and distribution of the staining. The expression of p16 protein is low in normal tissues, but high in abnormal tissues (*e.g.*, cervical intraepithelial neoplasia and cervical carcinoma) which stain with a brownish-yellow or yellow color.

## Statistical methods

Statistical analyses were performed using SPSS software (version 21.0; IBM Corp., Armonk, NY, USA). The consistency of the image parameter values measured by the two radiologists was examined by calculating the intraclass correlation coefficients (ICC) for APT, MK, and MD. The Shapiro–Wilk test was used to test the normality of the APT, MK, and MD values, and after the Chi-square test, one-way analysis of variance (ANOVA) was used to analyze the normally distributed data. The rank-sum test was used to analyze the non-normally distributed data, and ROC curves were plotted. The Delong test was used to compare the predictive efficacies of APT, MK, and MD values, both individually and in combination. The predictive efficacy of cervical cancer p16 was compared with APT, MK, and MD values alone and in combination by plotting the ROC curve and Delong test, and the area under ROC curve (AUC) was calculated, setting $0.5 \sim \leq 0.7$ as low diagnostic value, $0.7 \sim \leq 0.9$ as intermediate diagnostic value, and $>0.9$ as high diagnostic value. The sensitivity and specificity of the indicators were determined based on the Jordon index. Finally, the correlation between each parameter and cervical cancer p16 was analyzed using Spearman's correlation test.

# RESULTS

## Consistency of measurement results

The intragroup correlation coefficients of the values of each parameter measured by the two observers in this study were 0.882, 0.839, and 0.841 (ICC > 0.75), respectively, indicating very good agreement between the two observers. The mean values of the two measured parameters were averaged for subsequent analyses.

## Comparison of imaging parameters between p16 positive and negative groups

The APT, MK, and MD values of the patient groups with p16-positive and negative results were normally distributed. The APT, MD and MK values in the p16-positive group were $(3.179 \pm 0.518)\%$, $0.882 \pm 0.234 \ (\times 10^{-3}) \ \text{mm}^2/\text{s}$, and $1.024 \pm 0.297$, respectively; and in the p16-negative group, the APT, MD and MK values were $(2.828 \pm 0.542)\%$, $1.073 \pm 0.425 \ (\times 10^{-3}) \ \text{mm}^2/\text{s}$, and $0.846 \pm 0.359$, respectively. The APT and MK values in the p16-positive group were greater than those in the negative group. The MD values in the

**Table 2 Comparison of imaging parameters between p16 positive group and negative group.**

| Parameter | p16 positive ($n = 35$) | p16 negative ($n = 20$) | $p$ |
|---|---|---|---|
| APT mean (%) | 3.179 ± 0.518 | 2.828 ± 0.542 | <0.001 |
| MD (mm$^2$/s) | 0.882 ± 0.234 (×10$^{-3}$) | 1.073 ± 0.425 (×10$^{-3}$) | 0.001 |
| MK | 1.024 ± 0.297 | 0.846 ± 0.359 | <0.001 |

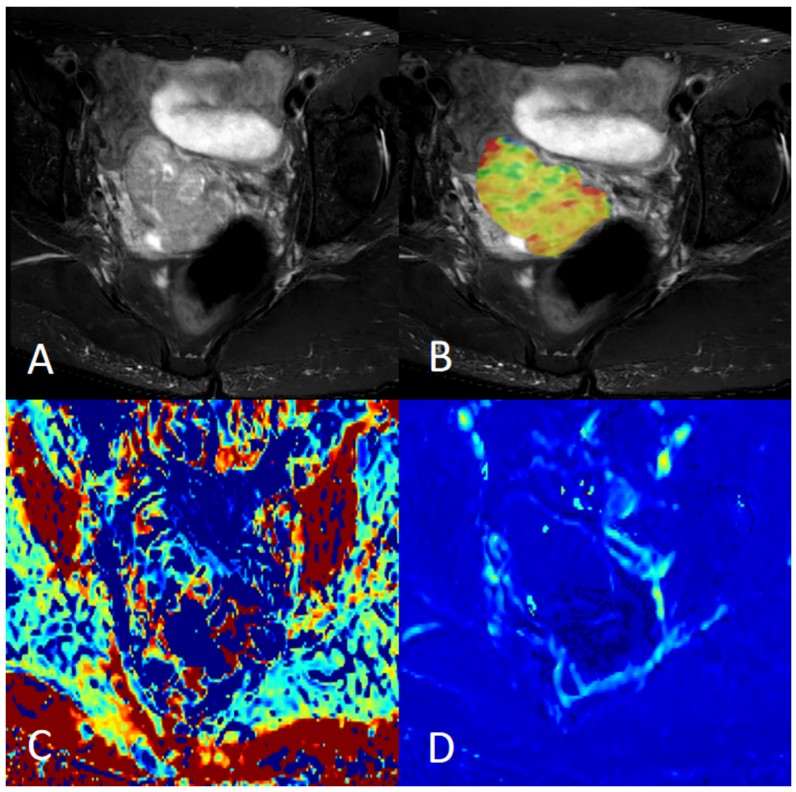

**Figure 1 Female, 58 years old, squamous cell carcinoma of the cervix, p16 positive.** (A) Axial fat saturation T2 weighted image; (B) fusion image of APT pseudo-colour image and fat saturation T2WI with APT value of 3.20%; (C) MK pseudo-colour image with MK value of 0.91; and (D) MD pseudo-colour image with MD value of $0.97 \times 10^{-3}$ mm$^2$/s.

p16-positive group were smaller than those in the negative group, and the differences were statistically significant ($P < 0.05$) (Table 2, Figs. 1, 2).

## Prediction of p16 expression by each parameter

The area under curve (AUC), sensitivity, and specificity of APT value, MK value, MD value, APT + MK value, APT + MD value, and APT + MK + MD value for predicting cervical cancer p16 were 0.809, 80.00%, and 75.0%; 0.801, 91.43%, and 70.0%; 0.790, 85.71%, and 70.0%; 0.845, 85.71%, and 80.0%; 0.866, 77.14%, and 90.0%; 0.871, 82.86%, and 90.0%, as shown in Table 3.

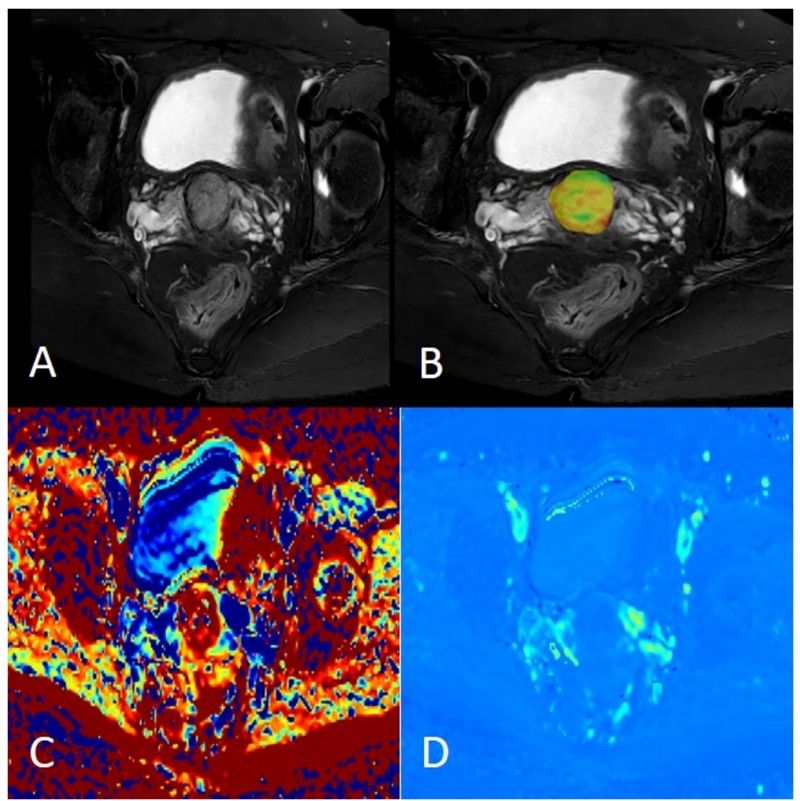

**Figure 2 Female, 38 years old, squamous cell carcinoma of the cervix, p16 negative.** (A) Axial fat saturation T2 weighted image; (B) fusion image of APT pseudo-colour image and fat saturation T2WI with APT value was 2.60%; (C) MK pseudo-colour image, MK value was 0.69; (D) MD pseudo-colour image, MD value was $1.26 \times 10^{-3}$ mm$^2$/s.

**Table 3 Evaluation of different combination models for predicting p16 in cervical cancer.**

| Combination model | AUC | 95% CI | Sensitivity | Specificity |
|---|---|---|---|---|
| APT mean | 0.809 | [0.681–0.903] | 80.00% | 75.0% |
| MK | 0.801 | [0.672–0.897] | 91.43% | 70.0% |
| MD | 0.790 | [0.659–0.888] | 85.71% | 70.0% |
| APT mean + MK | 0.845 | [0.722–0.928] | 85.71% | 80.0% |
| APT mean + MD | 0.866 | [0.747–0.943] | 77.14% | 90.0% |
| APT mean + MK + MD | 0.871 | [0.754–0.946] | 82.86% | 90.0% |

## Difference in the predictive efficacy of each parameter for cervical cancer p16

The comparative effects of APT, MK, and MD values alone and in combination, on the AUC for the prediction of cervical cancer p16 status were shown in Fig. 3. APT values in combination with MK and MD values predicted the largest AUC for cervical cancer p16. The order from high to low was APT + MK + MD value > APT + MD value > APT + MK value > APT value > MK value > MD value.

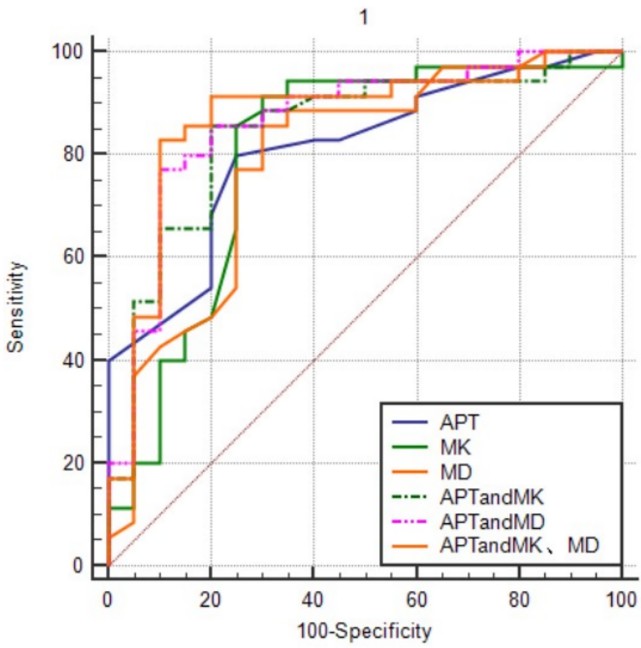

**Figure 3  The ROC curves of different combination models.** APT values in combination with MK and MD values predicted the largest AUC for cervical cancer p16.

## Correlation analysis of two imaging parameters with cervical cancer p16 expression

The APT and MK values were positively correlated with p16 expression in cervical cancer ($rs = 0.516$, $0.456$, $P < 0.01$), and the MD values were negatively correlated ($rs = -0.507$, $P < 0.01$).

## DISCUSSION

DKI, first proposed in *Jensen et al. (2005)*, is a new diffusion imaging technique based on the theory of non-Gaussian distribution of water molecules in tissues. DKI considers the complexity of the diffusion of water molecules through tissues, and introduces a fourth-order three-dimensional tensor to the original diffusion imaging model. This accurately quantifies the diffusion characteristics of water molecules in tissues, thus reflecting the complexity of tissue microstructure with high sensitivity (*Jensen & Helpern, 2010*). Inactivation of p16 function is often accompanied by severe genomic instability and an aggressive phenotype, resulting in disturbed cellular arrangement, increased nucleoplasmic ratios and necrotic areas. DKI is able to capture such changes in microstructural complexity by quantifying the non-Gaussian distribution properties of water molecule diffusion. Elevated MK values are often indicative of increased diffusion limitation (reflecting high cellular densities or membrane structural disturbances).

Unlike DKI, APT reveals lesion characteristics by detecting the content of free proteins and peptides in tissues (*Milot, 2022*). The principle of APT imaging is the saturation of amide protons in free proteins/peptides in tissues using specific RF pulses, and then the use

of the chemical exchange between the amide water protons to convert the barely detectable change observed in amide proton concentration into a detectable change in the water signal. This allows the indirect measurement of free proteins and peptides in tissues without the use of exogenous contrast. P16, a cell cycle regulatory protein, is overexpressed in response to the HPV oncogene E7, which leads to cell cycle deregulation and triggers aberrant proliferation of tumour cells by inhibiting the CDK4/6-Rb pathway. This proliferation process is accompanied by increased energy metabolic demands, such as enhanced glycolysis (Warburg effect) and accelerated protein synthesis, thus altering the acid-base balance of the tumour microenvironment. APT imaging can be used as a marker of metabolic abnormality because it can sensitively reflect changes in intracellular protein concentration and pH by detecting the chemical exchange effect of the amide proton with free water.

A recent study by *Li et al. (2023)* demonstrated the importance of APT imaging for determining the type and grade of cervical cancer, which improved the noninvasive prediction of prognostic factors. Similar conclusions were reached by *Meng et al. (2020)*, who concluded that both DKI and APT could be used for the initial assessment of cervical cancer. However, DKI was superior to APT in identifying pathological types, grading, and staging.

All patients with resectable cervical cancer (including early stage 1B cases and some stage 2A cases) are recommended to first undergo regional pelvic and para-aortic lymph node dissection, followed by histopathological examination to exclude microscopic lymph node metastasis (*Díaz-Feijoo et al., 2020*). If regional lymph nodes show invasion, radical hysterectomy should be avoided and the patient offered definitive radiation and chemotherapy (*Park et al., 2021*). Lymph node dissection is a life-threatening surgical procedure due to the risk of vascular injury and postoperative deep vein thrombosis (*Harter et al., 2019*). It is desirable to have a soft marker that can be detected at an early stage to replace operative lymph node dissection. Studies have suggested that p16 expression is strongly associated with prognosis (*McGrath et al., 2017*; *Chopra et al., 2025*). *El Sokkary & Sheta (2023)* corroborated this view in a recent study. In their study, they concluded that the rate of immunopositivity for p16 in squamous cell carcinoma of the uterine cervix was 56.7%, and that its positive staining was highly correlated with resectable early staging by clinical and imaging assessments. Therefore, it is important to further explore the correlation between multimodal magnetic resonance and p16 in cervical cancer.

In the present study, we compared the APT- and DKI-derived parameters individually and in combination. We found that the combination of APT and DKI was superior to either treatment alone in predicting p16 expression. APT had the highest AUC values for predicting p16 expression in cervical cancer when used together with MK and MD values. Although the intergroup differences predicted by the combination were not statistically significant, the different sequences provided different perspectives on the tissue microstructure, which contributed to a better assessment of tumor biology and improved diagnostic accuracy. Currently, studies on the prediction of p16 expression in cervical cancer using APT and DKI alone have not been reported; however, our results support the

possibility of such a prediction and suggest that the combined use of APT and DKI can achieve better prediction results.

This study had some limitations. First, the number of patients in this study was relatively small. Second, there is no uniform standard for the number and value of b-values used in DKI scanning, and the reproducibility of the b-values used in this study is uncertain. Third, the use of p16 as a biomarker has limitations. There is a false-positive rate for the expression of p16 protein in tissue cells, and the diagnostic method, which is mainly based on morphology, is greatly influenced by subjective factors of pathologists. Fourth, this study intentionally avoided cystic and necrotic regions when ROIs were depicted, which reduced the heterogeneity of tumor tissues. This may have affected the diagnostic value of certain parameters. Some studies have shown that it can improve the diagnostic specificity to screen, diagnose, and treat cervical precancerous lesions and cervical cancer through the combined detection of p16 protein with other tumor biomolecular markers or by adding the results of later follow-up. Therefore, in future, we will attempt to incorporate more tumor biomolecular markers to enable reliable and reproducible prediction of the results.

## CONCLUSIONS

APT, MK, and MD values help predict p16 expression status in cervical cancer before surgery, and the combined use of APT and DKI can more accurately predict p16 expression in cervical cancer than the application of either index alone. We accurately assessed the p16 status of patients with cervical cancer before surgery, which correlated closely with resectable disease, thereby potentially providing a minimally-invasive alternative to pelvic lymph node dissection for guiding clinical decision-making, optimizing treatment plans, and improving patient outcomes.

## ACKNOWLEDGEMENTS

We would like to thank Editage for English language editing.

### Funding

The authors received no funding for this work.

### Competing Interests

The authors declare that they have no competing interests.

### Author Contributions

- XingChen Wu conceived and designed the experiments, performed the experiments, analyzed the data, prepared figures and/or tables, authored or reviewed drafts of the article, and approved the final draft.
- Chen Xu performed the experiments, analyzed the data, prepared figures and/or tables, and approved the final draft.

- Xiaoyan Zhang performed the experiments, analyzed the data, prepared figures and/or tables, and approved the final draft.
- Qianqian Qu conceived and designed the experiments, performed the experiments, authored or reviewed drafts of the article, and approved the final draft.
- Zhe Wang performed the experiments, prepared figures and/or tables, and approved the final draft.
- Kai Deng conceived and designed the experiments, analyzed the data, authored or reviewed drafts of the article, and approved the final draft.

## Human Ethics

The following information was supplied relating to ethical approvals (*i.e.*, approving body and any reference numbers):

The First Affiliated Hospital of Shandong First Medical University & Shandong Provincial Qianfoshan Hospital granted Ethical approval to carry out the study within its facilities (Ethical Application Ref: 2021S849).

## Data Availability

The raw measurements are available in the Supplemental File.

## Supplemental Information

Supplemental information for this article can be found online at http://dx.doi.org/10.7717/peerj.19387#supplemental-information.

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
