# Peer review of "Preoperative prediction of p16 expression in cervical cancer by amide proton transfer imaging combined with diffusion kurtosis imaging: a retrospective study"

_PeerJ, doi:10.7717/peerj.19387_

## Round 0.1 · original submission · Major Revisions

The authors are requested to carefully revise the manuscript and answer the questions raised by the reviewers.

Reviewer 1 ·

Basic reporting

Major comments
1. The introduction and discussion were poorly organized.
1. Need language improvement.

Minor Comments
Title
1. The title need to be improved to clearly indicate the subject of the study
2. P16 should be with low case

Abstract
3. Results: It is hard to understand the sentence of "P16 positivity was highly correlated with resectabletumors when assessed clinically and radiologically.". How did you get this result?

Introduction
4. line 48: the author should give more explain to the sentence of "Cervical cancer is caused by several factors".
5. line 60-61: What is the role of p16 expression in clinical pratice for cervical cancer diagnosis and treatment?
6. line 66-69: How about the previous applications of APT and DKI in cervical cancer?

Experimental design

Methods
7. More details MRI parameters for APT and DKI should be provided (a table is recommended to list the parameters). The authors can refer to several previous papers (for examples, JMRI, 2023, 57(2): 493-505; Quant Imaging Med Surg 2023;13(12):8157-8172)
8. More details about the reconstruction of APT and DKI images should be provided.
9. Line111-112: Are the ROIs used for APT and DKI images same or not? How to keep the consistency of ROIs between APT and DKI?

Validity of the findings

Results
10. line 151-155: the detail APT, MD, and MK values should be provided.
11. Line 157-158: More details results by the ROC analysis should be discribed.
13. Line 163: did you do any analysis regarding the cervical cancer grading?

Discussion
14. Line168-172: these sentence should be moved to the introduction section.
15. the author should discuss more about the underlying mechanism for the correlation between APT/DKI parameters and the p16 expression in cervical cancer.

Additional comments

Figures and Tables
16. Figure 1: all the subplots should be with the same field of view.
17. Table 1: the number of digits should be same.

·

Basic reporting

It performs the correlation of APT/DKI parameter characteristics of preoperative MR functional sequences and the histochemical expression of P16 in cervical cancer to better inform treatment selection and to help determine prognosis.


The hypothesis formation is needed to establish the problem in a more systematic way.

Experimental design

1 The method needs to be described in more detail by establishing the hypothesis or with problem statement.

Validity of the findings

1. The figures need to provide more details by marking through allow and indicated in different comparative images.

2. The method can also be shown through the flowchart or block diagram.

3. The conclusion would be able to resolve the problem statement that you will provide in hypothesis.

Additional comments

1. The correlation can be verified in the further studies before using in the clinical application would be a good process. if the sufficient number of previous studies are available on this can be discussed in the discussion section.

---

## Round 0.2 · Minor Revisions

The authors are requested to carefully revise the manuscript and answer the questions raised by the reviewers.

Reviewer 1 ·

Basic reporting

The authors have addressed most of my previous review comments, but I have some more concern.

Experimental design

The details about APT and DKI image reconstruction are still unavailable. How were the parameter maps calculated, and what equations were used?

Validity of the findings

What are the units for the APT, MD and MK values?

Additional comments

No

·

Basic reporting

The recommended comments have been incorporated in the manuscript appropriately.

Experimental design

The recommended comments have been incorporated in the manuscript appropriately.

Validity of the findings

The recommended comments have been incorporated in the manuscript appropriately.

---

## Round 0.3 · accepted · Accept

After revisions, one reviewer agreed to publish the manuscript. There is one reviewer left with a minor revision, and I think the author has responded adequately. I also reviewed the manuscript and found no obvious risks to publication. Therefore, I also approved the publication of this manuscript.